# Auxiliary Learning Induced Graph Convolutional Networks

## Abstract

Graph convolutional networks (GCNs) have recently achieved great success in many applications. However they suffer from an incomplete annotation problem for complex graph-structured data. In this paper, we introduce a novel auxiliary learning method for GCNs in a multi-task fashion, which can efficiently enrich the data annotations. Specifically, both link prediction and label generation are used as two auxiliary tasks to complement the primary task of node classification. These two auxiliary tasks are jointly trained with the primary node classification task via a graph meta-learning strategy. The experimental results demonstrate that the proposed method consistently and significantly outperforms existing methods and achieves state-of-the-art results on several benchmark citation network datasets.

## 1 Introduction

Graph-structured data is ubiquitous in real-world applications. However, general deep learning methods, such as convolutional neural networks (CNNs), cannot adapt to graph-structured data directly, because the nodes in a graph have different numbers of neighbors, which often lose the ranking information. To handle the graph data effectively, graph convolutional networks (GCNs) have recently been proposed and used in many applications such as biomolecular prediction [1] and recommendation systems [2].

Previous methods focus on designing models that can extract information from both the graph topology and node features. Specifically, existing GCN methods typically design different propagation strategies [3, 4, 5] for each network layer and stack more network layers [6, 7, 8] to derive larger receptive fields. However, the neighborhood aggregation is essentially a type of Laplacian smoothing and stacking too many layers may result in over-smoothing [9]. These drawbacks of existing methods limit further performance enhancement.

In this paper, we try to explore the bottleneck of node classification from another point of view, i.e., from the training data itself. As shown in Fig. 1, graph-structured data has different properties from grid-like data such as an image. The most obvious difference is that the nodes in a graph are connected by edges. This causes two main issues whien it comes to annotating graph-structured data, resulting in that existing methods cannot fully leverage the graph-structured information. First, the edges in most graph data for semi-supervised node classification are unweighted. This arbitrary edge indication setting cannot effectively reflect the detailed graph structures. Besides, graph-structured data may be contaminated with noisy edges. These noisy edges cannot represent the true pairwise relationships between nodes. Second, using one-hot labels to train a graph-based model is inappropriate. One-hot labels are widely used in various machine learning tasks, assigning a training sample to a single class. However, nodes in a graph are connected; even nodes with different classes may have relations. In this scenario, it is more suitable to use soft labels to assign a node to multiple classes, with different probabilities indicating which class the node possibly belongs to.

Submitted to 35th Conference on Neural Information Processing Systems (NeurIPS 2021). Do not distribute.

To fully leverage the graph-structured information for enhancing the node classification performance of GCNs, we introduce an auxiliary learning scheme to the GCN framework in a multi-task fashion. To this end, we add two auxiliary tasks to enrich the topology information of a graph by softening the node labels and re-weighting the edges. Experimental results show that our model achieves state-of-the-art node classification performance on several benchmark citation network datasets. Our contributions are as follows:

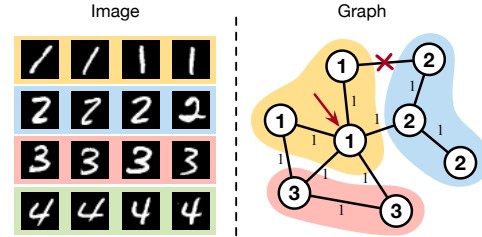

Figure 1: **The difference between image data and graph-structured data.** The nodes in graph-structured data are connected. Further, in most graph data for the node classification task, edges are unweighted and noisy.

(1) We propose two auxiliary tasks to capture more accurate graph information and enhance the model performance. The auxiliary link prediction task ensures that the model captures more graph topology information and generates probabilistic edges. The auxiliary label generation task softens the one-hot labels and generates pseudo-labels for unlabeled nodes.

(2) The reconstructed edges and pseudo sudo labels derived via the two auxiliary tasks are iteratively updated with a node classifier of the primary task based on a meta auxiliary learning strategy, resulting in state-of-the-art node classification performance.

## 2 Related Work

Over the past few years, GCNs have achieved significant breakthroughs in graph data representation. Generally, existing GCNs can be divided into spectral-based methods and spatial-based methods.

The spectral-based methods use graph spectral theory to define the graph convolutional operation in a graph Fourier domain. Spectral CNN [10] follows these mathematical foundations, assuming that a convolutional filter is a set of learnable parameters. To reduce computational complexity, ChebNet [11] approximates a graph convolutional filter as Chebyshev polynomials of the eigenvalues. GCN [12] introduces a first-order approximation of ChebNet and proposes a renormalization trick to alleviate numerical instabilities and exploding/vanishing gradients. DualGCN [13] introduces a dual GCN architecture with two graph convolutional layers in parallel to encode both local and global structural information.

The spatial-based methods define feature aggregation in the spatial domain directly, which is more efficient, general, and flexible [14]. The key challenge for these spatial-based methods is to apply the convolution operation for different-sized neighborhoods, while at the same time maintaining the weight sharing property. Neural network for graphs (NN4G) [15] is the first spatial-based method, applying the graph convolutional operation in the spatial space. Diffusion convolutional neural networks (DCNN) [16] consider graph convolutions as diffusion processes to efficiently learn features that are invariant under isomorphism. Message passing neural networks (MPNN) [17] model graph convolution as a message passing process among the nodes. The graph attention network (GAT) [3] introduce masked self-attentional layers to assign different weights to adjacent nodes, leading to learnable filter weights. The mixture model network (MoNet) [18] introduces pseudo-coordinates to assign different weights to the neighbors of each node. To achieve weight sharing across different nodes, some spatial-based models attempt to rank a node's neighbors via certain criteria or metrics, which transforms the graph-structured data into grid data for further processing. The large-scale graph convolutional network (LGCN) [19] ranks a node's neighbors via the node feature values. Then, multiple 1D convolutional layers are stacked for feature aggregation. Approximate personalized propagation of neural predictions (APPNP) [4] takes the personalized PageRank algorithm as the model propagation method to avoid over-smoothing when stacking more layers or increasing the size of the neighborhood.

Multi-task learning is designed to simultaneously learn a set of related but different tasks for ensuring that a learning model can derive the best performance across all tasks. Different from multi-task learning, auxiliary learning is only concerned with model performance on the primary task. For

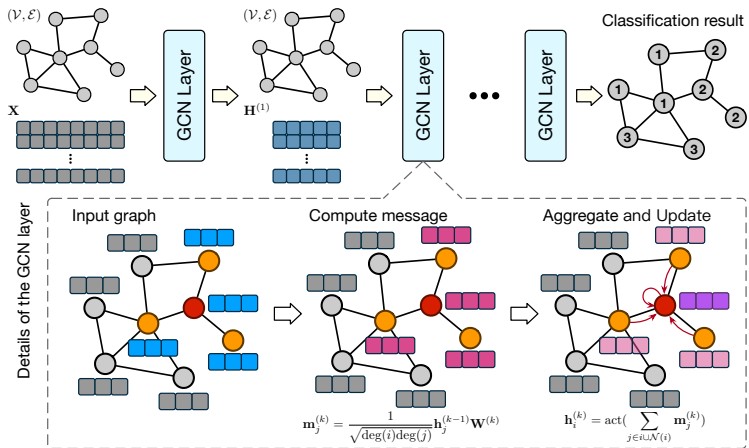

Figure 2: **Network architecture of the vanilla GCN model.** The vanilla GCN contains multiple GCN layers. Each layer captures the graph structure to generate the hidden embeddings from the previous layer (For the first layer, it is the original feature of the node) as input, and obtains the output through the message calculation, aggregation and update step. The last layer uses a softmax function to generate classification probabilities for each node.

instance, Deepstereo [20] leverages auxiliary learning to predict the relative poses of multiple cameras for unsupervised monocular depth estimation. To improve the performance of conversational speech recognition, auxiliary learning [21] is applied to low-level representations. Compared to the common learning scheme, meta auxiliary learning can enhance learning performance. For instance, MAXL [22] adopts meta-learning to automatically generate the auxiliary task labels. Pseudo Label [23] is a semi-supervised learning method, where a deep neural network is trained using both labeled and unlabeled data. For unlabeled data, the model picks up the class that has the maximum predicted probability as the true label to train itself. MPL [24] extends the Pseudo Label [23] via a meta-learning strategy, where the pseudo-labels are not generated by itself, but by a teacher network.

The existing GCNs are designed for a single task, where the properties of graph-structured data are not fully explored. We introduce the auxiliary learning scheme to leverage more detailed graph topology information for enhancing node classification performance.

## 3 Method

### 3.1 Preliminaries

Given a graph $\mathcal{G} = \{\mathcal{V}, \mathcal{E}, \mathbf{X}\}$, $\mathcal{V}$ is a set of nodes and $\mathcal{E}$ is a set of the edges connecting the related nodes. $\mathbf{X} \in \mathbb{R}^{N \times d}$ represents the features matrix of the nodes, where $d$ is the dimension of the node features and $N = |\mathcal{V}|$ is the number of nodes.

The proposed method adopts the vanilla GCN [12] as the backbone network, taking the graph adjacency matrix $\mathbf{A}$, labeled training set $\mathbf{Y}_{\text{train}}$, and original features $\mathbf{X}$ of the nodes as inputs to perform the semi-supervised node classification task. Based on the MPNN framework [17], each layer of the vanilla GCN is defined in three parts:

(1) *Message computation*: The message of node $v_i$ and its neighbor node $v_j$ is calculated, where $j \in \mathcal{N}(i)$, as:

$$\mathbf{m}_j^{(k)} = \frac{1}{\sqrt{\deg(i)\deg(j)}} \mathbf{h}_j^{(k-1)} \mathbf{W}^{(k)}, \; j \in \{i\} \bigcup \mathcal{N}(i). \tag{1}$$

Here, $\deg(i)$ is the degree of node $v_i$ and $\mathbf{W}^{(k)} \in \mathbb{R}^{c_{k-1} \times c_k}$ are the learnable parameters of the $k^{\text{th}}$-layer, where $c_k$ is the size of the hidden embedding.

(2) *Aggregation*: The messages of node $v_i$ and its neighbors are aggregated by summing them up:

$$\mathbf{m}_{\mathcal{N}(i)}^{(k)} = \sum_{j \in \{i\} \cup \mathcal{N}(i)} \mathbf{m}_j^{(k)}, \tag{2}$$

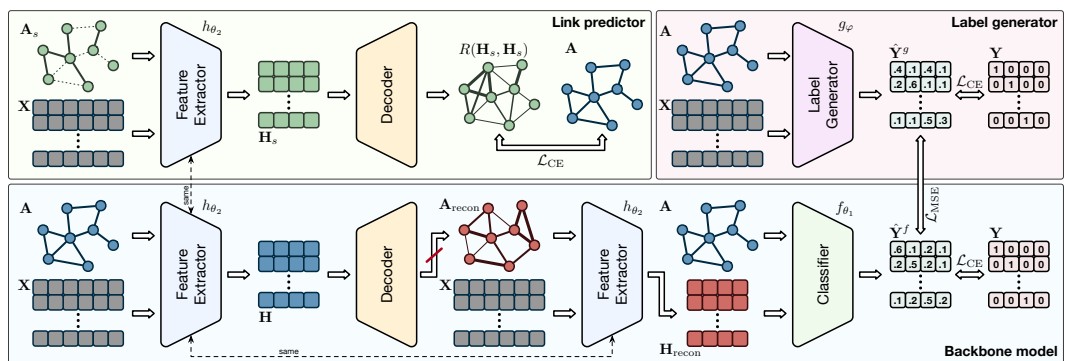

Figure 3: **The overall network architecture.** Our method consists of three networks: a backbone network for the primary task and two auxiliary task networks. (1) The backbone network is a vanilla GCN, which predicts the classification results of each node. (2) The first auxiliary task network is a link predictor, which focuses on the link prediction task and generates a probabilistic edge structure as the input of the backbone network. (3) The second auxiliary task network is a label generator, which employs the label generation task to generate the pseudo soft labels for supervising the node classifier. The parts connected by the dashed arrow share the same parameters, and the red slash on the arrow indicates a stop-gradient (`detach`) operation.

where $\mathbf{m}_{\mathcal{N}(i)}^{(k)}$ denotes the aggregated message.

(3) *Feature updating*: Finally, the hidden representation is updated with the aggregated message. For the vanilla GCN, the *update* function can be considered as applying a non-linear operation to the aggregated message:

$$\mathbf{h}_i^k = \begin{cases} \mathrm{softmax}(\mathbf{m}_{\mathcal{N}(i)}^{(k)}), & \text{if } k = K \\ \mathrm{ReLU}(\mathbf{m}_{\mathcal{N}(i)}^{(k)}), & \text{otherwise.} \end{cases} \tag{3}$$

Here, $K$ is the number of model layers. The last layer of the vanilla GCN should output the classification probability via a softmax function. Otherwise, a ReLU operation is used.

## 3.2 Multi-Task Network Architecture

In this paper, we propose an auxiliary learning induced GCN for semi-supervised node classification (see Fig. 3). To enhance the node classification performance of the backbone vanilla GCN, we design two auxiliary tasks: link prediction and pseudo-label generation.

The first $K-1$ layers of a $K$-layer vanilla GCN model can be considered as a feature extractor, while the last layer can be considered as a classifier. Denoting the feature extractor and node classifier of the backbone model as $h_{\theta_2}$ and $f_{\theta_1}$, respectively, where $\theta_1$ and $\theta_2$ are the learnable parameters, the proposed two auxiliary task networks are defined as follows.

### 3.2.1 Link Predictor

To enrich the edge information, we design an auxiliary link prediction task to infer the missing edges and present the probability of edge existence. The link predictor we propose contains a decoder $R(\cdot)$ and the feature extractor $h_{\theta_2}$ of the backbone model.

The feature extractor $h_{\theta_2}(\cdot)$ takes reduced adjacency matrix $\mathbf{A}_s$ which corresponds to the sampled edge set $\mathcal{E}_s \subset \mathcal{E}$, and node features $\mathbf{X}$ as inputs to generate the hidden embedding:

$$\mathbf{H}_s = h_{\theta_2}(\mathbf{X}, \mathbf{A}_s). \tag{4}$$

Then, the decoder computes the similarity between each node based on $\mathbf{H}_s$ to predict the edge existence probabilities. We simply use an inner-product calculator with a sigmoid function as the implementation of decoder $R(\cdot)$. The similarity of two hidden embeddings can be computed as

$$r_{ij} = R(\mathbf{h}_i, \mathbf{h}_j) = \sigma(\mathbf{h}_i \mathbf{h}_j^T), \tag{5}$$

where $\mathbf{h}_i$ and $\mathbf{h}_j$ denote the hidden embeddings of node $v_i$ and node $v_j$ in $\mathbf{H}_s$, respectively.

### 3.2.2 Label Generator

Although using the one-hot label is inappropriate, it is difficult to obtain soft labels with manual annotations for real graph-structured data. To tackle this issue, we introduce an auxiliary label generation task to generate soft labels that reflect the tendency of different classes each node belongs to. Label generator $g_\varphi(\mathbf{X}, \mathbf{A})$ is a vanilla GCN, where $\varphi$ are the learnable parameters and $\mathbf{A}$ is the adjacency matrix. The label generation network predicts the label distribution of each node based on the graph structure $\mathbf{A}$ and the raw features $\mathbf{X}$ of the nodes, as:

$$\hat{\mathbf{Y}}^g = g_\varphi(\mathbf{X}, \mathbf{A}), \tag{6}$$

where $\mathbf{Y}^g$ are the predicted pseudo labels for guiding the training of the backbone network and the label generator.

### 3.2.3 Node classifier

The node classifier carries out the primary semi-supervised node classification task in our model. Compared to the vanilla GCN model, we add a graph reconstruction step at the beginning. The graph adjacency matrix reconstructed with the hidden embeddings contains richer edge information since the proposed auxiliary link prediction task can enhance the graph topology capturing ability of the feature extractor $h_{\theta_2}(\cdot)$. However, the hidden embeddings derived by the feature extractor change rapidly in the first few iterations, resulting in a changing reconstructed adjacency matrix. Directly applying the reconstructed adjacency matrix to the entire backbone network increases the training instability. Thus, we only apply it as the input of the feature extractor $h_{\theta_2}(\cdot)$, while the classifier $f_{\theta_1}(\cdot)$ still adopts the original adjacency matrix as input.

In each training iteration, the feature extractor $h_{\theta_2}(\cdot)$ first generates the hidden embeddings of the nodes $\mathbf{H} = h_{\theta_2}(\mathbf{X}, \mathbf{A})$. The decoder uses $\mathbf{H}$ to reconstruct the adjacency matrix $\mathbf{A}_{\mathrm{recon}} = R(\mathbf{H}, \mathbf{H})$. Then, the feature extractor takes the reconstructed adjacency matrix $\mathbf{A}_{\mathrm{recon}}$ to compute the hidden embeddings $\mathbf{H}_{\mathrm{recon}} = h_{\theta_2}(\mathbf{X}, \mathbf{A}_{\mathrm{recon}})$.

Finally, the computed hidden embeddings $\mathbf{H}_{\mathrm{recon}}$ and the original graph adjacency matrix $\mathbf{A}$ are fed to the classifier to obtain the final classification results:

$$\hat{\mathbf{Y}}^f = f_{\theta_1}(\mathbf{H}_{\mathrm{recon}}, \mathbf{A}). \tag{7}$$

## 3.3 Auxiliary Training Phases

In addition to the backbone network, the proposed method contains two auxiliary task networks. In the following, we will introduce the objectives for the node classifier $f_{\theta_1}$, link predictor $h_{\theta_2}$, and label generator $g_\varphi$ in order, and then leverage a meta auxiliary learning scheme to train the proposed multi-task network.

### 3.3.1 Training the Node Classifier $f_{\theta_1}$

The purpose of the node classifier $f_{\theta_1}(\cdot)$ is to carry out the graph-based semi-supervised node classification task, which yields the final prediction results. Naturally, it is necessary to use the classification loss between the predicted result and the real categories of nodes to supervise the training process. At the same time, to reflect the tendency of the class a node belongs to, the training should make the prediction labels of the classifier be close to the pseudo soft labels generated by label generator $g_\varphi(\cdot)$.

In the $t^{\mathrm{th}}$ iteration, denoting the pseudo soft labels as $\hat{\mathbf{Y}}^{g(t)} = g_{\varphi^{(t)}}(\mathbf{X}, \mathbf{A})$, the real labels used in training as $\mathbf{Y}_{\mathrm{train}}$, and the prediction results of the node classifier of the backbone network $f_{\theta_1}$ as $\hat{\mathbf{Y}}^{f(t)} = f_{\theta_1^{(t)}}(h_{\theta_2^{(t)}}(\mathbf{X}, \mathbf{A}_{\mathrm{recon}}^{(t)}), \mathbf{A})$, the objective for the node classifier $f_{\theta_1}(\cdot)$ is defined as

$$\mathcal{L}_{\theta_1}^{(t)} = \mathcal{L}_{\mathrm{CE}}(\hat{\mathbf{Y}}_{\mathrm{train}}^{f(t)}, \mathbf{Y}_{\mathrm{train}}) + \mathcal{L}_{\mathrm{MSE}}(\hat{\mathbf{Y}}^{f(t)}, \hat{\mathbf{Y}}^{g(t)}). \tag{8}$$

The objective contains two parts: the loss on the real training labels, and the loss on the generated pseudo soft labels. $\mathcal{L}_{\mathrm{CE}}$ denotes the cross-entropy loss and $\mathcal{L}_{\mathrm{MSE}}$ denotes the mean squared loss. Although this objective can be used to update the learnable parameters $\theta_2$ of the feature extractor $h_{\theta_2}(\cdot)$, as it generates the hidden embeddings used for node classification, we only employ it to

supervise the learning of the node parameters $\theta_1$ of the classifier $f_{\theta_1}(\cdot)$. To avoid adding unnecessary supervision for generating the reconstructed graph adjacency matrix $\mathbf{A}_{\text{recon}}$, we add a stop-gradient operation (detach) which is formulated as follows:

$$\mathbf{A}_{\text{recon}}^{(t)} = \texttt{detach}(R(\mathbf{H}, \mathbf{H})). \tag{9}$$

### 3.3.2 Auxiliary Training for the Feature Extractor $h_{\theta_2}$

The feature extractor $h_{\theta_2}$ is shared by the link prediction module and the backbone network. In each iteration, we first randomly sample a certain percentage of edges from the real edge set $\mathcal{E}$ to form a sampled edge set $\mathcal{E}_s \subset \mathcal{E}$. Denoting the reduced graph adjacency matrix as $\mathbf{A}_s$, which corresponds to the sampled edge set $\mathcal{E}_s$, and applying a message passing operation with $\mathbf{A}_s$ as

$$\mathbf{H}_s^{(t)} = h_{\theta_2^{(t)}}(\mathbf{X}, \mathbf{A}_s^{(t)}), \tag{10}$$

then the objective for the feature extractor is defined as

$$\mathcal{L}_{\theta_2}^{(t)} = \mathcal{L}_{\text{CE}}(R(\mathbf{H}_s^{(t)}, \mathbf{H}_s^{(t)}), \mathbf{A}). \tag{11}$$

Here $R(\mathbf{H}_s^{(t)}, \mathbf{H}_s^{(t)})$ represents the correlation between the hidden embeddings of each pair of nodes.

### 3.3.3 Auxiliary Training for the Label Generator $g_\varphi$

The label generator $g_\varphi$ is a vanilla GCN model used to predict the label distribution for each node, which naturally needs to be supervised by a classification loss. In the $t^{\text{th}}$ iteration, a label generator first generates the prediction results using the original graph adjacency matrix $\mathbf{A}$ and the node features $\mathbf{X}$, which is formulated as $\hat{\mathbf{Y}}^{g(t)} = g_{\varphi^{(t)}}(\mathbf{X}, \mathbf{A})$. Denoting the training labels as $\mathbf{Y}_{\text{train}}$, the objective of the label generator $g_\varphi$ can be formulated as

$$\mathcal{L}_\varphi^{(t)} = \mathcal{L}_{\text{CE}}(\hat{\mathbf{Y}}_{\text{train}}^{g(t)}, \mathbf{Y}_{\text{train}}). \tag{12}$$

### 3.3.4 Meta-Learning Based Training Strategy

The final node classification results only depend on the primary task, while the performance of the other modules, including the label generator and link predictor are not our ultimate concern. Simply using the classification loss to train the label generator $g_\varphi$ or using the link prediction loss to train the feature extractor $h_{\theta_2}$ cannot provide the effects of auxiliary learning. Thus, it is crucial to make the node classifier $f_{\theta_1}$ perform better after it is trained with the pseudo soft labels while taking the hidden embeddings derived by the feature extractor $h_{\theta_2}$ as inputs. To this end, we use meta auxiliary learning to update the model parameters.

For the auxiliary label generation task, we consider not only the classification performance of the label generator $g_\varphi$, but also the auxiliary effect on the node classifier. We assume that the classifier parameters $\theta_1$ are updated with gradient descent based on the pseudo soft labels in the $t^{\text{th}}$ iteration,

$$\theta_1' = \theta_1^{(t)} - \eta \nabla_{\theta_1} \mathcal{L}_{\text{MSE}}(\hat{\mathbf{Y}}^{f(t)}, \hat{\mathbf{Y}}^{g(t)}). \tag{13}$$

A direct way to evaluate the auxiliary effect of the pseudo soft labels is to compute the classification loss of the prediction given by the node classifier using the updated parameters $\theta_1'$, as

$$\mathcal{L}^{f'} = \mathcal{L}_{\text{CE}}(f_{\theta_1'}(h_{\theta_2^{(t)}}(\mathbf{X}, \mathbf{A}_{\text{recon}}), \mathbf{A})_{\text{train}}, \mathbf{Y}_{\text{train}}). \tag{14}$$

This loss can quantify how much performance improvement the classifier gains from the two auxiliary tasks. Because $\theta_1'$ is updated with the pseudo soft labels generated by $g_\varphi(\cdot)$, the loss $\mathcal{L}^{f'}$ is also a function of $\varphi$. This means that the objective could be used to supervise the learning of $\varphi$. Note that $\nabla_\varphi \mathcal{L}^{f'}$ requires the gradient of the gradient to be computed [25], which can be considered as a meta-learning strategy. The final objective of the label generator $g_\varphi$ with meta-learning is formulated as

$$\mathcal{L}_{\varphi-\text{meta}}^{(t)} = \mathcal{L}_\varphi^{(t)} + \mathcal{L}^{f'}. \tag{15}$$

Similar to the label generator $g_\varphi(\cdot)$, the link predictor $h_{\theta_2}^b$ should derive the effective node embeddings to enhance the node classification performance. Since the hidden embeddings used for node

**Algorithm 1** AL-GCN
___
**Input:** Graph adjacency matrix $\mathbf{A}$, the node features $\mathbf{X}$, the data labels $\mathbf{Y}_{\text{train}}$ of a training set.
**Output:** A feature extractor $h_{\theta_2}$, a node classifier $f_{\theta_1}$
1: Initialize learnable parameters $\theta_1$, $\theta_2$, $\varphi$
2: **while** *not converged* **do**
3:      # *node classifier training phase*
4:        $\hat{\mathbf{Y}}^g \leftarrow g_\varphi(\mathbf{X}, \mathbf{A})$
5:        $\mathbf{H} \leftarrow h_{\theta_2}(\mathbf{X}, \mathbf{A})$
6:        $\mathbf{A}_{\text{recon}} = \texttt{detach}(R(\mathbf{H}, \mathbf{H}))$
7:        $\mathbf{H}_{\text{recon}} \leftarrow h_{\theta_2}(\mathbf{X}, \mathbf{A}_{\text{recon}})$
8:        $\hat{\mathbf{Y}}^f \leftarrow f_{\theta_1}(\mathbf{H}_{\text{recon}}, \mathbf{A})$
9:        $\mathcal{L}_{\theta_1} \leftarrow \mathcal{L}_{\text{CE}}(\hat{\mathbf{Y}}^f_{\text{train}}, \mathbf{Y}_{\text{train}}) + \mathcal{L}_{\text{MSE}}(\hat{\mathbf{Y}}^f, \hat{\mathbf{Y}}^g)$
10:       Update: $\theta_1 \leftarrow \text{Adam}(\mathcal{L}_{\theta_1}, \theta_1)$
11:      # *meta-learning preparation*
12:       Compute: $\theta'_1 \leftarrow \theta_1 - \eta \nabla_{\theta_1} \mathcal{L}_{\text{MSE}}(\hat{\mathbf{Y}}^f, \hat{\mathbf{Y}}^g)$
13:       $\mathcal{L}^{f\prime}$ $\leftarrow \mathcal{L}_{\text{CE}}(f^b_{\theta'_1}(h^b_{\theta_2}(\mathbf{X}, \mathbf{A}_{\text{recon}}), \mathbf{A})_{\text{train}}, \mathbf{Y}_{\text{train}})$
14:      # *label generator training phase*
15:       $\hat{\mathbf{Y}}^g \leftarrow g_\varphi(\mathbf{X}, \mathbf{A})$
16:       $\mathcal{L}_\varphi \leftarrow \mathcal{L}_{\text{CE}}(\hat{\mathbf{Y}}^g_{\text{train}}, \mathbf{Y}_{\text{train}}) +$ $\mathcal{L}^{f\prime}$
17:       Update: $\varphi \leftarrow \text{Adam}(\mathcal{L}_\varphi, \varphi)$
18:      # *feature extractor training phase*
19:       $\mathbf{A}_s \leftarrow \text{RandomSample}(\mathbf{A})$
20:       $\mathbf{H}_s \leftarrow h_{\theta_2}(\mathbf{X}, \mathbf{A}_s)$
21:       $\mathcal{L}_{\theta_2} \leftarrow \mathcal{L}_{\text{CE}}(R(\mathbf{H}_s, \mathbf{H}_s), \mathbf{A}) +$ $\mathcal{L}^{f\prime}$
22:       Update: $\theta_2 \leftarrow \text{Adam}(\mathcal{L}_{\theta_2}, \theta_2)$
23: **end while**
___

classification are derived from the feature extractor $h_{\theta_2}$, the objective defined in Eq. 14 can also be considered as a function of $\theta_2$. Thus, the objective of the feature extractor $h_{\theta_2}$ with meta-learning is defined as

$$\mathcal{L}^{(t)}_{\theta_2-\text{meta}} = \mathcal{L}^{(t)}_{\theta_2} + \mathcal{L}^{f\prime}. \tag{16}$$

This objective means that, except for the link prediction task, the feature extractor $h_{\theta_2}$ should ensure that the generated hidden embedding enhance the classification accuracy of the node classifier $f_{\theta_1}$ updated with the pseudo soft labels. The overall training process is shown in Alg. 1.

# 4 Experiments

## 4.1 Experimental Settings and Compared Methods

We demonstrate the classification performance of our method via semi-supervised document classification on three citation network datasets, including Cora, Citeseer, and Pubmed [26], where nodes represent the documents and edges are citation links. Dataset statistics are summarized in Table 1.

The proposed method is a novel GCN, which is leveraged to carry out a node classification task. We compare our method with several popular graph-based node classification methods including GCN [12], GAT [3], DualGCN [13], SGC [27], and APPNP [4]. As we use a three-layer GCN as the backbone in our proposed model, we compare both two-layer and three-layer GCNs, denoted as GCN2 and GCN3, respectively.

For the semi-supervised node classification, we use all node features but only 20 labels per class for training and 500 nodes as the validation set. We train the proposed method for a maximum of 400 epochs using Adam [28] with a learning rate of 0.01. We use the well-trained parameters, which achieve the best performance on the validation set during the training phases, to evaluate classification accuracy on a test set of 1,000 labeled examples. We run each method 100 times and compute the average classification accuracy on a single NVIDIA GTX 1080Ti GPU. To implement all the compared methods more conveniently, we use PyG [29] as the graph-based learning framework.

Table 1: Dataset statistics.

| Dataset | Nodes | Edges | Features | Classes |
|---------|-------|-------|----------|---------|
| Cora | 2,708 | 5,278 | 1,433 | 7 |
| CiteSeer | 3,327 | 4,552 | 3,703 | 6 |
| Cora | 19,717 | 44,324 | 500 | 3 |

Table 2: Classification results on the datasets (**bold**: best, underline: runner-up).

| Method | Cora | Citeseer | Pubmed |
|--------|------|----------|--------|
| GAT [3] | $82.5 \pm 0.8\%$ | $71.4 \pm 0.7\%$ | $78.4 \pm 0.4\%$ |
| DualGCN [13] | $83.4 \pm 0.5\%$ | **$72.6 \pm 0.6\%$** | $79.9 \pm 0.3\%$ |
| SGC [27] | $81.3 \pm 0.7\%$ | $70.9 \pm 0.6\%$ | $78.2 \pm 0.5\%$ |
| APPNP [4] | $83.2 \pm 0.4\%$ | $71.6 \pm 0.5\%$ | $79.8 \pm 0.3\%$ |
| GCN2 [12] | $81.5 \pm 0.7\%$ | $71.5 \pm 0.5\%$ | $79.2 \pm 0.4\%$ |
| **AL-GCN2 (ours)** | $82.3 \pm 0.4\%$ | **$72.6 \pm 0.5\%$** | $79.6 \pm 0.5\%$ |
| GCN3 [12] | $80.7 \pm 1.2\%$ | $68.0 \pm 1.4\%$ | $77.7 \pm 0.5\%$ |
| **AL-GCN3 (ours)** | **$84.7 \pm 0.4\%$** | $72.3 \pm 0.5\%$ | **$81.4 \pm 0.6\%$** |

## 4.2 Experimental Results

In this section, we provide the experimental results of the node classification, an ablation study, and the visualization of hidden embeddings. More experimental results, such as parameter and model robustness studies can be found in the supplementary materials.

We conduct the node classification task on three citation network datasets. As shown in Table 2, our method consistently and significantly enhances the learning performance compared to the other methods. In particular, for the Cora dataset, the proposed method is superior to GCN by 4.9%. Compared to the other methods, our model considers more graph-structured information via auxiliary learning. Thus, it is consistently and significantly superior to the compared methods, achieving state-of-the-art results.

## 4.3 Ablation Studies

### 4.3.1 On the Auxiliary Learning Modules

To determine how the link predictor ($P$) and the pseudo label generator ($G$) affect the node classification performance, we apply the following two ablation models: (1) Vanilla GCN with the link predictor, termed GCN+$P$. (2) Vanilla GCN with the lable generator, termed GCN+$G$. We compare GCN+$G$ and GCN+$P$ with the proposed method, AL-GCN (GCN+$P$+$G$), and the original GCN. Table 3 lists the results. As can be seen, the proposed method consistently outperforms GCN+$P$ and GCN+$G$. Specifically, compared to the link predictor, the label generator has more effect on the Citeseer dataset. However, for the Pubmed dataset, the link predictor has much more effect on the learning performance.

### 4.3.2 On the Reconstructed Graph Adjacency Matrix

For the backbone network, a reconstructed graph adjacency matrix via a link prediction task is used as input. To determine how the reconstructed graph adjacency matrix affects the node classification performance, we compare the proposed model with the following two models: (1) A model that takes the original graph adjacency matrix without the reconstructed one as the input of the backbone network, termed w/o-recG; (2) A model that takes only the reconstructed graph adjacency matrix as the input of the backbone network, termed w/o-oriG.

As shown in Table 4, our method consistently outperforms w/o-recG and w/o-oriG. This can be attributed to the fact that the reconstructed graph adjacency matrix via the link predictor can capture the detailed topology information of a graph and the fixed original graph adjacency matrix increases the training stability.

Table 3: The ablation experiment results.

| Method | Cora | Citeseer | Pubmed |
|---|---|---|---|
| GCN | $80.7 \pm 1.2\%$ | $68.0 \pm 1.4\%$ | $77.7 \pm 0.5\%$ |
| GCN+$P$ | $83.0 \pm 0.7\%$ | $70.6 \pm 0.8\%$ | $81.3 \pm 0.6\%$ |
| GCN+$G$ | $83.0 \pm 0.6\%$ | $71.7 \pm 0.8\%$ | $78.8 \pm 0.6\%$ |
| **GCN+$P$+$G$ (AL-GCN)** | $84.7 \pm 0.4\%$ | $72.3 \pm 0.5\%$ | $81.4 \pm 0.6\%$ |

Table 4: The ablation experiment results in terms of classification accuracy (in percent).

| Method | Cora | Citeseer | Pubmed |
|---|---|---|---|
| GCN | $80.7 \pm 1.2\%$ | $68.0 \pm 1.4\%$ | $77.7 \pm 0.5\%$ |
| w/o-recG | $84.5 \pm 0.5\%$ | $71.9 \pm 0.6\%$ | $80.1 \pm 0.5\%$ |
| w/o-oriG | $84.1 \pm 0.6\%$ | $71.4 \pm 0.9\%$ | $80.6 \pm 1.5\%$ |
| **AL-GCN** | $84.7 \pm 0.4\%$ | $72.3 \pm 0.5\%$ | $81.4 \pm 0.6\%$ |

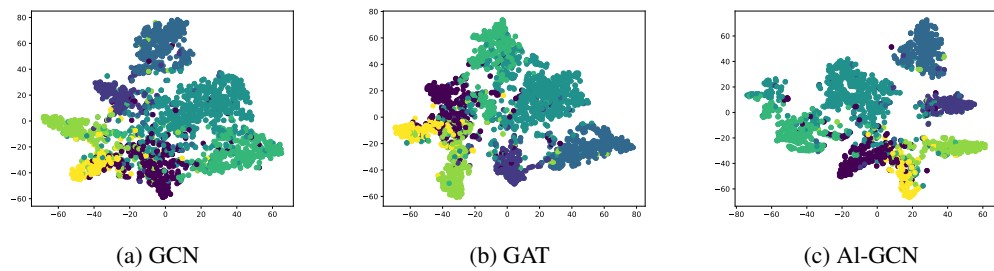

(a) GCN       (b) GAT       (c) Al-GCN

Figure 4: Visualization of the hidden embedding obtained by different methods via t-SNE algorithm.

## 4.4 Visualization of Hidden Embeddings

To determine how the hidden embeddings affect the learning performance, we use the visualization tool t-SNE [30] to observe their distribution. As shown in Fig. 4, the embedding results of GCN and GAT are denser, and the separation of different clusters is not obvious. In contrast, the node distributions learned by our proposed method are more separate, with most of the nodes from the same classes being close to each other, resulting in obvious cluster structures. These experimental results demonstrate that the proposed method can capture more detailed structure information of a graph, including the nodes and edges, resulting in more effective hidden embeddings.

## 5 Conclusion

We have proposed a novel graph convolutional network for semi-supervised node classification. Different from existing methods, the proposed model focuses on enriching the graph data and adopts meta auxiliary learning to enhance the representations of nodes and edges in a graph. To enrich node label information, an auxiliary label generator is used to generate pseudo probabilistic labels. Meanwhile, an auxiliary link predictor is used to generate probabilistic edges to enrich the graph structure information. The enriched node and edge information can iteratively enhance the performance of the node classification task. Experimental results on several benchmark citation datasets show that the proposed model is superior to the existing methods. For future work, we note that real-world data is usually contaminated by noise, which results in a robustness problem for graph learning methods. We plan to extend our model to handle noisy data by designing a more robust learning method for graph-structured data.

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
