# OpenReview forum: "Auxiliary learning induced graph convolutional networks"
_NeurIPS.cc/2021/Conference — NeurIPS 2021 Submitted_

### Official Review · Reviewer_KGBu · 2021-07-02

**Rating:** 6
**Confidence:** 3

**Summary:**

In this paper, they introduced a method for node classification in graphs using Graph Convolutional Networks (GCNs) in multi-task fashion to address incomplete annotation problem. Two auxiliary tasks, link prediction and label generation, are trained with the target task (node classification) simultaneously using a graph meta-learning approach.


**Ethical Concerns:**

I have not observed any ethical issues in this work.

**Limitations And Societal Impact:**

As far as I understand, for the limitations, they mentioned that they plan to address the noise in the data by designing more robust method.

They did not mentioned any potential negative societal impact in the work; and I do not see there could be any negative societal impact in this work as well.

**Main Review:**




Strengths:

- In general, paper is well-written and well-organized; e.g.,  Algorithm 1 is self-descriptive. There are some small confusion for me that I mention later.

-  I like the idea of using two related auxiliary tasks to improve node classification problem; specifically, they used two link prediction and label generation tasks, in which they leveraged graph-structured information to improve the performance of target task (node classification).

- The experiments are thorough, they cover both qualitative and qualitative analysis, the ablation studies were interesting

Notes:

- Figure 1 is a little bit confusing for me. I would suggest using the citation network example instead of image example in the figure 1 to be consistent with the experiment data.

- For figure 2 and 3, I would recommend adding explanation related to color-coded features and nodes.for example, what green and blue colored nodes mean? it will be also better if they could add some explanation in the captions of tables.

- In the link prediction task, if I understand it correctly, they used the similarity of every two node to compute if edge is between them? What is the threshold to infer there is an edge between two nodes?

- In the link prediction task, it would be useful to mention the intuition why they use a sample of edges for each iteration? Is it related to time or complexity issues?

- In equation 8, for calculating the loss, I think for the psudo labels, you could use the cross entropy loss instead of MSE. What is the intuition of using MSE?

- In section 4.3.2, I do not understand the difference between "w/o-oriG" and "AL-GCN". My assumption is that both of them use the reconstructed adjacency matrix for backbone model. What is the difference between these two?

- In comparison with the other node classification methods, I would recommend adding a new paraphrase explaining why we should use "AL-GCN" instead of other methods; basically what is the winner characteristics of this method comparing others. Does this have better time complexity? Or does it perform better in semi-supervised setting? I understand it outperforms previous ones with respect to accuracy, but we could argue other benefits of using this approach

- I think this framework could be model-agnostic. I means we will be able to use GATs instead of of GCN? This will be nice future work.

- In line 56, one extra "sudo"?

**update: After receiving authors' response, I would like to appreciate their time for addressing my concerns. My evaluation on the paper is reflected after reading their response.


**Time Spent Reviewing:**

8

---

> ### Author Response · Authors · 2021-08-09
> **Response to Reviewer KGBu**
>
> Thank you for the comments. For each of your questions, we will answer them separately below.
>
> **Q1:** Figure 1 is a little bit confusing for me. I would suggest using the citation network example instead of image example in the figure 1 to be consistent with the experiment data.
>
> **A1:** For Figure 1, we only abstract a few nodes to represent some ubiquitous characteristics of a graph structure.
>
> **Q2:** For figure 2 and 3, I would recommend adding explanation related to color-coded features and nodes. For example, what green and blue colored nodes mean? it will be also better if they could add some explanation in the captions of tables.
>
> **A2:** For Figure 2 and 3, considering that the limitation of the space, some descriptions are omitted. For Fig. 2, different colors represent different types (e.g., the red node represents the center node and the nodes in orange are the neighbors). For Fig. 3, we hope that different colors can distinguish different graphs and their corresponding generated feature representations (e.g., the graph in green represent the down-sampled graph and the features in green represent the feature extracted with it).
>
> **Q3:** In the link prediction task, if I understand it correctly, they used the similarity of every two node to compute if edge is between them? What is the threshold to infer there is an edge between two nodes?
>
> **A3:** Because the link prediction task is an auxiliary task and its result is not what we finally concern, we did not binarize the existence probabilities of edges based on the threshold. However, we choose a part of the edges through different thresholds \tau, and directly use the predicted probability as the weight of the edge. In this way, the weights of the edges are re-adjusted. Generally, the predicted probability of an edge is mapped to the weight of [-1,1], and we reserve edges with a weight greater than 0. The specific parameter analysis is shown in the supplementary material.
>
> **Q4:** In the link prediction task, it would be useful to mention the intuition why they use a sample of edges for each iteration? Is it related to time or complexity issues?
>
> **A4:** Yes, it is related to the training time. It is very time-consuming to calculate the probability between all node pairs in each iteration. Moreover, it is also related to training stability. The parameters of the model are initialized randomly. If we calculate the similarity between all node pairs at the beginning, it is very easy to produce wrong results and affect the stability of training.  Adding a part of node pairs to it in each iteration can avoid the introduction of noise edges at the beginning of training phases.
>
> **Q5:** In equation 8, for calculating the loss, I think for the pseudo labels, you could use the cross entropy loss instead of MSE. What is the intuition of using MSE?
>
> **A5:** The CE loss can indeed be used, but the CE loss is suitable for one hot label in the real implementation (like pytorch). Besides, CE loss is not asymmetric, and our model hopes that this loss can be used for the meta-learning step of the label generation network, so that the symmetric loss is better.
>
> **Q6:** In section 4.3.2, I do not understand the difference between "w/o-oriG" and "AL-GCN". My assumption is that both of them use the reconstructed adjacency matrix for backbone model. What is the difference between these two?
>
> **A6:** AL-GCN only uses the reconstructed graph on each layer of feature extractor and does not use a reconstructed graph on the classifier to ensure the training stability. However, w/o-oriG uses the reconstruction graph on every layer of the backbone model. It can be seen that the instability of the reconstructed graph at the beginning of training affects the entire training process, resulting in slightly lower performance.
>
> **Q7:** In comparison with the other node classification methods, I would recommend adding a new paraphrase explaining why we should use "AL-GCN" instead of other methods; basically what is the winner characteristics of this method comparing others. Does this have better time complexity? Or does it perform better in semi-supervised setting? I understand it outperforms previous ones with respect to accuracy, but we could argue other benefits of using this approach.
>
> **A7:** The reason for using AL-GCN: It can directly refine the structure and label annotation of a graph to improve the performance. As shown in the supplementary material, our method is more robust on noise edge experiment. For the time complexity, our method does not have advantages compared to Vanilla GCN, because meta-learning is more time-consuming.
>
> **Q8:** I think this framework could be model-agnostic. I mean we will be able to use GATs instead of GCN? This will be nice future work.
>
> **A8:** Yes, it can be used on several different graph neural networks. This will be the future work.
>
> **Q9:** In line 56, one extra "sudo"?
>
> **A9:** It is a mistake. Thank you for the comments. We will carefully proofread the entire paper.

---

### Official Review · Reviewer_rGes · 2021-07-15

**Rating:** 6
**Confidence:** 4

**Summary:**

This paper proposes a novel meta-learning framework that utilizes auxiliary learning tasks of link predictions and label generations to improve the performance of GCNs on node classifications tasks. The meta-learning algorithm adopts the idea from [Model-Agnostic Meta-Learning for Fast Adaptation of Deep Networks](https://arxiv.org/abs/1703.03400) so that the models for auxiliary tasks are updated in a way that has higher impacts on the primary task.  The authors use two self-supervised learning tasks, edge predictions and label generation, as the auxiliary tasks.  The method performs favorably in the three standard public graph data sets with higher prediction accuracy.

**Limitations And Societal Impact:**

The limitations and potential negative sociteal impacts are not addressed.

**Main Review:**

This paper is generally well written and well structured. I appreciate the authors' efforts to keep the discussions organized and easy to understand. The experiment's setting is reasonable and the results of the experiment suggest that the proposed method is able to achieve higher prediction accuracy. And the design choices are properly examined in the ablation study.

Using self-supervised tasks as auxiliary learning tasks in GNNs has been discussed in a few pieces of literature recently, such as [1][2]. Different from prior arts, this paper presents a novel meta-learning algorithm to the GNNs that using the impacts on the primary tasks to direct the update of the auxiliary tasks. Thus the primary tasks might suffer less from degeneration, which is common for multi-task learning.  However, I am curious about the difference between this method and the one in [3], which is general auxiliary learning using self-supervised tasks and a similar meta-learning. In general, I find the contribution of this paper quite incremental.


[1] Multi-Stage Self-Supervised Learning for Graph Convolutional Networks on Graphs with Few Labeled Nodes

[2] When Does Self-Supervision Help Graph Convolutional Networks?

[3] Self-Supervised Generalisation with Meta Auxiliary Learning, NeurIPS 2019.


**Time Spent Reviewing:**

6

---

> ### Author Response · Authors · 2021-08-09
> **Response to Reviewer rGes**
>
> Thank you for your review and affirmation of our work.
>
> **Q1:** I am curious about the difference between this method and the one in [3], which is general auxiliary learning using self-supervised tasks and a similar meta-learning. In general, I find the contribution of this paper quite incremental.
>
> **A1:** Similar to Self-Supervised Generalisation with Meta Auxiliary Learning (MAXL), our method is used auxiliary tasks and meta-learning to improve the performance of the model. However, our method has the following key differences:
> 1. MAXL only generates sub-category labels on the training data that already has the labels. However, the proposed method is to generate pseudo-labels for all training data in the semi-supervised task, which can be regarded as softening and amplifying the labels according to the graph structure.
> 2. The training of MAXL's label generator is completely dependent on the gradient obtained by the meta-learning strategy. However, the proposed method does not rely on meta-learning, and the improved performance can still be achieved without the meta-learning strategy (Please refer to supplementary materials).
> 3. We employed MAXL on GCN with 3 sub-categories for each real category. We observe that the training process is very unstable and reduces the performance of the backbone network, which is: Cora: 78.6±1.7%, Citeseer: 64.2±2.4%, Pubmed:76.3±1.5%.

---

### Official Review · Reviewer_t7yf · 2021-07-17

**Rating:** 4
**Confidence:** 4

**Summary:**

The paper proposes a new auxiliary learning method to improve the existing GCNs. Considering that the edges in many graphs are unweighted and nodes labels are often one-hot, the authors added two auxiliary tasks to enrich the topology information in the graph node classification task. One is link prediction which generates probabilistic edges; and the other is to generate soft labels. The probabilistic edges and soft labels are iteratively updated by a meta auxiliary learning strategy in a smart way, and the results on three standard benchmark datasets outperform some baselines.


**Limitations And Societal Impact:**


I do not see any negative social impact.

**Main Review:**

The motivation of the paper is reasonable but maybe not so significant. It is limited to the node classification task on graphs with binary edges. In practice many graphs have weights or attributes on edges, and the application of most GNNs are also not limited to the node classification task.

The paper combines the ideas of auxiliary learning and meta pseudo labels to enrich the topology information. It is novel in the context of graph learning, but some concepts or descriptions are confusing. For example, I am not sure whether the label generation can be regarded as an auxiliary task, since g_\phi is a totally different model which does not share any parameters with the target model in the main task.

The experiments are weak. As the area of deep graph learning is developing so fast, just using the three small graph datasets with standard settings are not enough nowadays. It cannot give us enough insights why and how the proposed method works. For example, 1. the advantage of using probability edges instead of original ones may be making the model not only more expressive but also robust. Can you make some perturbations on the edges and see how the performance changes? 2. The utilization of teacher model and pseudo labels maybe make the unlabeled data more efficiently used, so can you change the proportion of labeled/unlabeld data  to show the power of your model? Or can you experiment on larger datasets such as OGB?
This kind of experiments will make the paper more solid. Otherwise, it is not convincing.


**Time Spent Reviewing:**

5

---

> ### Author Response · Authors · 2021-08-09
> **Response to Reviewer t7yf**
>
> Thank you for reviewing the manuscript and for your valuable comments.
>
> **Q1:** The motivation of the paper is reasonable but maybe not so significant. It is limited to the node classification task on graphs with binary edges. In practice many graphs have weights or attributes on edges, and the application of most GNNs are also not limited to the node classification task.
>
> **A1:** We admit that this work is mainly for the semi-supervised node classification task of a graph. However, we just take it as an example to show the effect of the graph structure data itself and propose an auxiliary learning induced training scheme. Since the proposed method modifies the weights of edges according to the graph structure and node features, it is not only suitable for graphs with binary edges. Note that although the edges in some graphs have weights and other information, there is no guarantee that these weights are correct (e.g., click farming on e-commerce platforms). In this case, our method is still applicable.
>
> **Q2:** I am not sure whether the label generation can be regarded as an auxiliary task, since $g_\phi$ is a totally different model which does not share any parameters with the target model in the main task.
>
> **A2:** Why the label generation can be regarded as an auxiliary task, here we give the direct replies. Our method not only performed on the ground truth lables, but also the generated pseudo-labels, so the learning on the pseudo-label can be considered as an auxiliary task for the backbone model. Furthermore, the parameters of $g_\phi$ are updated via meta-learning strategy that depends on the backbone network. Although the label generator is a completely different model, its gradient update is closely related to the backbone model. Thus, we consider that the label generation can be regarded as an auxiliary task.
>
> **Q3:** The experiments are weak. As the area of deep graph learning is developing so fast, just using the three small graph datasets with standard settings are not enough nowadays. It cannot give us enough insights why and how the proposed method works.
>
> **A3:** We follow the practice of most papers and choose the commonly used three benchmark citation datasets with the public split. This paper is an exploration of data annotation and graph structure improvement, using these three simple datasets can basically demonstrate the effectiveness of the proposed method. To further explore the influence of graph structure, we conducted the experiments on noise edges. After gradually increasing the noise of the edges in a graph, the relative performance degradation of our method is the least (Please refer to the supplementary material). The experimental results demonstrate the proposed method improves the robustness of graph neural network. We will add the experiments you suggested in the final version.

---

### Decision · Program_Chairs · 2021-09-27

**Decision:**

Reject

**Comment:**

The paper has merit and received mixed reviews, but overall it is too incremental. I think there are several avenues along which the paper can be strengthened, some of which are mentioned in the discussion. Pursuing those, it seems likely to get to a paper with critical mass for a top tier ML conference. The submitted paper is not quite there yet.